# Adaptation Strategies of Seedling Root Response to Nitrogen and Phosphorus Addition

**DOI:** 10.3390/plants13040536

**Published:** 2024-02-15

**Authors:** Xing Jin, Jing Zhu, Xin Wei, Qianru Xiao, Jingyu Xiao, Lan Jiang, Daowei Xu, Caixia Shen, Jinfu Liu, Zhongsheng He

**Affiliations:** 1College of Forestry, Fujian Agriculture and Forestry University, Fuzhou 350002, China; fjnldxjinxing@163.com (X.J.); hdly0718@126.com (J.Z.); weiixn@163.com (X.W.); xiaoqianruchi@163.com (Q.X.); 18854885671@163.com (J.X.); jln0629@163.com (L.J.); xudaowei2004446@126.com (D.X.); 2Key Laboratory of Ecology and Resource Statistics in Fujian Province, Fuzhou 350002, China; 3School of Economics and Management, Sanming University, Sanming 365000, China; 13328581155@163.com

**Keywords:** root traits, resource-use strategy, nitrogen addition, phosphorus addition, *Castanopsis kawakamii*

## Abstract

The escalation of global nitrogen deposition levels has heightened the inhibitory impact of phosphorus limitation on plant growth in subtropical forests. Plant roots area particularly sensitive tissue to nitrogen and phosphorus elements. Changes in the morphological characteristics of plant roots signify alterations in adaptive strategies. However, our understanding of resource-use strategies of roots in this environment remains limited. In this study, we conducted a 10-month experiment at the *Castanopsis kawakamii* Nature Reserve to evaluate the response of traits of seedling roots (such as specific root length, average diameter, nitrogen content, and phosphorus content) to nitrogen and phosphorus addition. The aim was to reveal the adaptation strategies of roots in different nitrogen and phosphorus addition concentrations. The results showed that: (1) The single phosphorus and nitrogen–phosphorus interaction addition increased the specific root length, surface area, and root phosphorus content. In addition, single nitrogen addition promotes an increase in the average root diameter. (2) Non-nitrogen phosphorus addition and single nitrogen addition tended to adopt a conservative resource-use strategy to maintain growth under low phosphorus conditions. (3) Under the single phosphorus addition and interactive addition of phosphorus and nitrogen, the roots adopted an acquisitive resource-use strategy to obtain more available phosphorus resources. Accordingly, the adaptation strategy of seedling roots can be regulated by adding appropriate concentrations of nitrogen or phosphorus, thereby promoting the natural regeneration of subtropical forests.

## 1. Introduction

Seedling is a critical stage in the natural regeneration process of forests and the most sensitive period to external environmental factors [1]. The successful settlement of seedlings during this stage significantly impact the natural regeneration and community succession of forests [2]. Plant roots are an important organ for seedlings to absorb nutrients and water resources, and their strong ecological plasticity provides significant support for plants to adapt to soil environments and withstand potential stresses [3]. Research has demonstrated that soil factors exert a significant influence on the root turnover process, encompassing growth and eventual death [4]. In response to changes in the soil environment, plants dynamically adjust their morphological structure and physiological activities, as evidenced by studies conducted by Kou et al. [5] and Adams et al. [6]. However, there are still many unresolved mysteries regarding how plant roots perceive and respond to changes under different soil nutrient conditions [7]. Therefore, further investigation is needed to elucidate the specific mechanisms and factors that influence this adjustment process.

The root economic spectrum (RES) describes a trade-off relationship among root traits, studying a shift of plant strategies from conservative and acquisitive resource-use strategies in response to environmental conditions [8,9,10,11,12,13]. For instance, plant species with acquisitive resource-use strategies usually enhance root length and root nitrogen content, which inevitably decreases root diameter [8,9,10]. On the other hand, plants with conservative resource strategies tend to increase the average diameter of their roots while decreasing the specific root length, focusing more on nutrient storage and transport [10,12]. Therefore, the functional traits composed of the morphological structure and physiological–biochemical characteristics of root tissue express the trade-off relationship between maximizing resource acquisition and minimizing construction costs in plants [14,15]. Additionally, underground traits (such as the roots) may be coordinated with aboveground traits (leaves and stems) in multiple dimensions, collectively influencing plant growth [16]. However, compared to the aboveground traits, the resource-use strategy for root tissue is more covert and complex [17]. The concealment and complexity associated with this phenomenon present significant challenges to our understanding of how root tissues adapt to soil nutrients. Thus, the elucidation of resource-use strategies employed by plant roots in diverse environments holds great significance in comprehending the adaptive mechanisms of plant growth.

Nitrogen (N) and phosphorus (P) are essential nutrient elements for plant growth, and individually or collectively limit the survival and regeneration of plant seedlings in terrestrial ecosystems. In recent years, global nitrogen deposition has led to continuous soil acidification in forests and nutrient imbalance, thereby reducing the absorption of plants for insoluble salts and exacerbating the limitation of soil phosphorus on plant growth [18,19,20]. With the intensification of nitrogen deposition, the accumulation of nitrogen elements in the soil can cause nitrogen-favoring species to grow rapidly in communities, enhancing their relative competitive advantage over other species and leading to changes in community structure [21]. Additionally, the long-term negative effect of reduced nitrogen forms (ammonia and ammonium) hinders the development of roots and buds. The soil-mediated acidification effect increases the loss of alkaline ions, increases the concentration of potentially toxic metals (such as AI^3+^), and reduces nitrification, leading to the accumulation of litter. The secondary stress and interference factors generated from this can also reduce individual vitality and increase herbivorous activity [22]. In summary, these factors may collectively lead to a decrease in community species diversity. Therefore, excessive nitrogen input exacerbates the inhibitory effect of soil phosphorus limitation on root growth, which is unfavorable for roots to acquire soil resources [23]. Sprengel and Justus von Liebig’s law of the minimum has indicated that plant growth will be determined by the scarcest resources (limiting factors) [24]. According to this law, excessive nitrogen input leads to a scarcity of phosphorus, which becomes a bottleneck for plant growth. However, plants have also evolved corresponding strategies through long-term adaptation. For example, in forest soils with low nitrogen content, plants promote nitrogen uptake by increasing the root surface area [25]. Additionally, plants with higher specific root lengths can enhance nutrient uptake capacity, while a higher root tissue density improves stress resistance, thus maintaining the high growth demands of plants [8]. The study by Schleuss et al. [26] found that nitrogen addition alleviates the limitation of phosphorus on plant growth in forests, and the interaction between nitrogen and phosphorus determines the survival strategies of plants in acquiring nutrients from forest soil. Moderate proportions of nitrogen and phosphorus additions can regulate the allocation of aboveground and belowground resources in seedlings and improve their stress resistance by enhancing the plasticity of plant root and leaf traits [27]. It has been shown that fine root biomass and root branching number decrease with the increase of soil nitrogen availability [28,29], and different plant species with different life habits also respond differently to nitrogen and phosphorus nutrient additions [29,30]. Therefore, a thorough understanding of the response characteristics of seedling roots to nitrogen and phosphorus nutrients is an important way to explain species resource-use strategies [31].

The subtropical region of China has long been affected by nitrogen deposition, resulting in soil nitrogen enrichment and phosphorus deficiency, which has become a major limiting factor for plant growth and development in subtropical forests [32,33]. At the same time, the imbalance of soil’s nitrogen-to-phosphorus ratio directly affects the phosphorus cycle in forest ecosystems, indirectly altering the ecosystem structure and function [23]. The *Castanopsis kawakamii* Nature Reserve in Sanming City, Fujian Province, China, has rich species diversity and exhibits the typical appearance of subtropical evergreen broad-leaved forests [34]. Currently, the diversity of plant communities in the region expresses a decreasing trend due to the limitation of phosphorus in forest soil, but whether seedlings have the same sensitivity to soil nitrogen and phosphorus elements remains poorly understood. As previously mentioned, nitrogen deposition can induce a series of processes that can result in a decline in the diversity of community species [22]. This effect is particularly noteworthy in acidic soils in subtropical regions of China [3]. In such soils, phosphorus tends to persist in an insoluble form through adsorption and fixation with substances like iron, aluminum, and soil clay [5]. Consequently, the conversion of inorganic phosphorus into insoluble phosphorus exacerbates the restriction of soil phosphorus on plant growth [3,5]. As a dominant tree species in this region, *C. kawakamii* is also a rare and endangered evergreen broad-leaved tall tree unique to the southern edge of China’s central subtropical region, with a narrow natural distribution range [35]. The uncertainty of seedling regeneration currently leads to a decrease in the conversion rate to saplings, and under conditions of limited resources, they exhibit weaker competitive ability [36]. Therefore, how to protect rare tree species within the protected area has become a concerning problem.

This study focused on *C. kawakamii* seedlings. Experiments were conducted with different nitrogen or phosphorus addition concentrations and nitrogen–phosphorus interactive addition concentrations to evaluate the response of seedling roots to these nutrient additions. Specifically, we addressed the following two questions: (1) Do the root traits of *C. kawakamii* seedlings conform to the theory of root economic spectrum (RES)? We hypothesized that the root economic spectrum is applicable to the changes in root traits of *C. kawakamii* seedlings, and that the roots of wild seedlings adopt a conservative resource-use strategy. (2) Has the addition of nitrogen and phosphorus changed the resource-use strategies of seedling roots? We hypothesized that phosphorus addition promotes a shift in root resource-use strategies from conservatism to acquisition, while nitrogen addition expresses a more conservative resource-use strategies.

## 2. Results

### 2.1. Root Morphology Traits in Response to Nitrogen and Phosphorus Addition

The interaction of nitrogen and phosphorus addition at different concentrations had significant differences in the effects on root traits (Figure 1 and Appendix A). Compared with the N_0_P_0_ treatment, the N_0_P_i_ treatment increased the specific root length (Figure 1a), whereas the root average diameter decreased (Figure 1b). The N_i_P_0_ promoted the accumulation of root biomass (Figure 1d), and root average diameter (Figure 1b), while decreasing the specific root length (Figure 1a). Overall, specific root length showed a decreasing trend with the increasing nitrogen or phosphorus concentration in the nitrogen–phosphorus interaction treatments, reaching the highest value in the N_1_P_1_ treatment (Figure 1a). Root average diameter expressed an increasing trend with the increasing nitrogen concentration in the nitrogen–phosphorus interaction treatments (Figure 1b), and showed a significant decreasing trend with the increasing phosphorus concentration in the N_3_P_i_ treatment. Root tissue density reached the maximum value in N_3_P_2_ treatment (Figure 1c). Root biomass showed an increasing trend with the increasing nitrogen concentration in the nitrogen–phosphorus interaction treatments, followed by a decreasing trend (Figure 1d). Furthermore, the root surface area and root branching number showed a similar trend to the specific root length (Appendix A), and they had a very significant correlation with the specific root length (Appendix A).

### 2.2. Response of Root Nitrogen and Phosphorus Content to Nitrogen and Phosphorus Addition

Different nitrogen and phosphorus addition treatments significantly influenced the nitrogen and phosphorus content of seedling roots (Figure 2a,b). The root nitrogen content responded positively to single phosphorus addition, single nitrogen addition, and nitrogen–phosphorus interaction, and reached a maximum value in the N_3_P_2_ treatment (Figure 2a). The single nitrogen and phosphorus addition had a significant positive effect on the root phosphorus content, and a maximum value occurred in the N_0_P_3_ treatment (Figure 2b).

### 2.3. Resource-Use Strategies of Root Traits in Different Nitrogen and Phosphorus Addition

The principal component analysis (PCA) of the root traits of *C. kawakamii* seedlings (Figure 3) showed that Dim1 and Dim2 explained 29.8% and 18.8% of the overall indicator changes, respectively, and the total cumulative variance explained 48.6%. As the phosphorus concentration increases with a single application of phosphorus (N_0_P_i_), the resource-use strategies of the root shift from conservative resource strategy (high root average diameter) to acquisitive resource-use strategies (high specific root length). On the other hand, as the nitrogen concentration of a single nitrogen application (N_i_P_0_) increases, the roots strategy shifts from an acquisitive resource-use strategy to a conservative strategy (Figure 3 and Appendix A). In addition, under the interaction of nitrogen and phosphorus addition, as the phosphorus concentration increases, the root strategy of low nitrogen and phosphorus interaction and medium nitrogen and phosphorus interaction changes from conservative resource strategy to acquisitive resource-use strategies (Figure 3 and Appendix A). However, the treatments with no nitrogen and phosphorus addition and high nitrogen phosphorus interaction are on the conservative side, and the resource-use strategy is relatively conservative (Figure 3 and Appendix A). Furthermore, Appendix A presents the principal component analysis of various root traits under different nitrogen and phosphorus addition treatments.

### 2.4. Comprehensive Evaluation of Nitrogen and Phosphorus Addition on Root Traits

Using Euclidean distance (K-means) to measure the differences in root traits of seedlings under different nitrogen and phosphorus concentrations, the shortest distance method was employed to cluster the 16 treatments based on the similarity of 8 root traits. Treatments with the highest similarity were clustered together, resulting in three groups (Figure 4). Among them, the roots of N_2_P_1_, N_1_P_3_, and N_0_P_3_ treatments exhibited higher specific root length, surface area, branch number, and lower average diameter, indicating the best promotion of root elongation and growth, and were thus classified into one group. The effects of N_1_P_2_, N_2_P_2_, N_0_P_1_, and N_1_P_1_ treatments on root traits were secondary to N_2_P_1_, N_1_P_3_, and N_0_P_3_ treatments, and were therefore classified into another group. The remaining treatments had the poorest impact on root growth and were classified into another group.

## 3. Discussion

### 3.1. Effects of Single Nitrogen and Phosphorus Addition on Root Traits

The nitrogen (N) and phosphorus (P) supply levels not only significantly affect the nitrogen and phosphorus content of the roots [37], but also result in different morphological responses as the nutrient gradient changes [38]. The results indicated that compared to no nitrogen and phosphorus addition, single phosphorus addition significantly increases the root nitrogen content and root phosphorus content (Figure 2a,b). With increasing phosphorus concentration, the root phosphorus content increases, while the root nitrogen content decreases. This indicates that phosphorus addition can enhance the nutrient content of the roots, which is consistent with the findings of Zhu et al. [39]. Moreover, the specific root length, root surface area, and root branching number also increase with increasing phosphorus concentration (Figure 1a and Appendix A), while the root average diameter and root tissue density show a decreasing trend (Figure 1b,c). This may be attributed to phosphorus addition alleviating the pressure of low phosphorus stress on root growth by stimulating root elongation and growth, resulting in more branching and a larger surface area for better absorption of soil nutrients [40]. However, other studies have shown that phosphorus addition can increase root productivity and root diameter, but has no impact on specific root length and root tissue density [41]. These discrepancies in the response of root growth to nitrogen (N) and phosphorus (P) additions observed between different plant species can likely be attributed to variations in experimental conditions and soil types [7]. Our research area is characterized by subtropical evergreen broad-leaved forests, which are known to be relatively deficient in phosphorus [7,33,35]. In contrast, the study area of the latter’s research focuses on tropical lowland forests, which are typically more fertile in terms of phosphorus. Furthermore, the type of phosphorus fertilizer used in their study (superphosphate) differs from the one used in our research (sodium dihydrogen phosphate) [41]. These differences in soil composition and phosphorus fertilizer application can contribute to the varying responses observed in root growth between the two studies.

In addition, single nitrogen addition increased root average diameter, root biomass, root nitrogen content, and root phosphorus content (Figure 1b,d and Figure 2a,b). Previous studies have shown that nitrogen addition significantly increases the acquisition of N and P, enhances photosynthesis in plant leaves, and leads to plants allocating more photosynthetic products to the roots to obtain more phosphorus to balance the relatively high nitrogen in the plant [42,43]. Our findings align with those of previous studies, which demonstrate that the addition of nitrogen facilitates passive absorption by the root system. This process leads to an increase in root diameter, allowing for greater nutrient storage to meet the demands of passive absorption [43,44]. In contrast, when nitrogen was applied as a single dose, it resulted in a decrease in specific root length and the number of root branches (Figure 1a and Appendix A). This could be attributed to alterations in root allocation strategies triggered by excessive nitrogen saturation [43]. Previous research has demonstrated that an increase in soil nutrient availability often prompts plants to reduce their investment in root nutrient uptake and allocate more resources towards above-ground growth [44]. These findings suggest that the proliferation and growth of the root system can be regulated by manipulating nutrient inputs, but the precise mechanisms underlying this regulation require further investigation [7].

Most studies have indicated that nitrogen deposition-induced soil acidification has negative effects on root health and nutrient uptake [45,46]. Although nitrogen addition increased root nitrogen content, root phosphorus content, and root biomass, the lower specific root length would to some extent reduce the ability of the roots to actively obtain soil resources [44]. This is possibly due to the severe nitrogen–phosphorus imbalance in the subtropical region, where phosphorus is already deficient [47], and nitrogen addition exacerbates the restriction of low phosphorus stress on root growth, which hinders the acquisition of soil resources [8]. In summary, single phosphorus addition plays an important role in alleviating low phosphorus stress in subtropical soils and promoting rapid acquisition of soil resources by seedlings [14,48]. The inhibitory effect of nitrogen addition on the root growth of seedlings contrasts strongly with the promoting effect of phosphorus addition [32,33], which supports our first hypothesis.

### 3.2. Effects of Nitrogen and Phosphorus Interactive Addition on Root Traits

Compared to the single addition of nitrogen or phosphorus, the combined effect of nitrogen and phosphorus stimulating root growth may be better than the separate addition of nitrogen or phosphorus [42,48]. The results indicated that compared to the nitrogen–phosphorus addition, the nitrogen–phosphorus interaction treatment significantly increased the specific root length, root nitrogen content, root phosphorus content, root surface area, and root branching number (Figure 1a, Figure 2a,b, and Appendix A), but decreased root average diameter and root tissue density (Figure 1b,c). Moreover, at the same concentration level, the nitrogen–phosphorus interaction resulted in higher root surface area and specific root length compared to single nitrogen or phosphorus additions, while the opposite was observed for root average diameter and root tissue density. This indicates that nitrogen–phosphorus interaction not only significantly promotes the growth, elongation, and branching differentiation of seedling roots but also has a greater promoting effect compared to the same level of single nitrogen or phosphorus addition. This finding, namely the synergistic effect resulting from the combined addition of nitrogen and phosphorus, has been supported by previous studies [26,43]. However, the precise mechanisms underlying these synergistic effects remain poorly understood. Schleiss et al. [26] propose that this mechanism can be explained by the enhanced accumulation of organic phosphorus in the soil due to the combination of nitrogen and phosphorus, as well as the stimulation of organic phosphorus hydrolysis through an increase in phosphatase activity. Furthermore, the combined addition of nitrogen and phosphorus has been shown to increase the relative abundance of arbuscular mycorrhizal fungi (AMF) genes and enhance phosphorus uptake [26].

Furthermore, the specific root length, root surface area, root branching number, and root biomass showed an initial increase and then a decreasing trend with increasing phosphorus concentrations under the nitrogen–phosphorus interaction treatment, and the decreasing trend was more pronounced in the high nitrogen and phosphorus interaction (Figure 1a,d and Appendix A). This suggests that the nitrogen–phosphorus interaction promotes root morphological growth [43,49]. In addition, the roots under the high nitrogen and phosphorus interaction treatment had higher root nitrogen content (Figure 2a), and the specific root length and root surface area showed a decreasing trend with the increasing phosphorus concentration (Figure 1a and Appendix A). This may be because the excessive nitrogen addition increases the available nitrogen content in the soil, enhances the nitrogen acquisition of roots, and thus increases the root nitrogen content [43,50]. To balance the higher nitrogen content in the roots, plants need to acquire more phosphorus to meet the demand. Therefore, excessive nitrogen can lead to an increased demand for phosphorus [51], and the higher nitrogen content in the roots can enhance phosphorus acquisition by increasing specific root length and root surface area [52]. However, with the increase in phosphorus concentration, this effect is to some extent alleviated. Hence, the smaller specific root length and root surface area can meet the requirements of root phosphorus [53].

### 3.3. Trade-Off Strategies of Root in Response to Nitrogen and Phosphorus Addition

To improve nutrient use efficiency and optimize growth, plants enhance the adjustment of root morphology as an important adaptive strategy [41,54]. The root economic spectrum (RES) provides a clear path for explaining the underground trade-off strategies of root traits [55]. With further research, it has expanded from the one-dimensional root economic spectrum (usually represented by the negative correlation between root nitrogen content and root tissue density [9,12]) to the multidimensional root economic spectrum (where the first dimension is primarily represented by root diameter and specific root length, and the second dimension represents the coordinated variations of root nutrients (i.e., N, P) and root tissue density [10,15]. These diverse combinations of root strategies can better explain the adaptive strategies of plants in more heterogeneous soil environments [11,12,13].

The results indicate that under the treatment of no nitrogen and phosphorus addition, single nitrogen addition, and high nitrogen and phosphorus interactive addition (N_0_P_0_, N_i_P_0,_ and N_3_P_i_), a conservative resource strategy was adopted by the roots (Figure 3). This suggests that root growth is restricted and not conducive to the acquisition of soil resources in these two situations. However, with the increase in phosphorus concentration under the treatment of high nitrogen and phosphorus interactive addition, the growth pressure on the roots was somewhat relieved [50]. In contrast, under the treatment of single phosphorus addition, low nitrogen and medium nitrogen, and phosphorus interactive addition (N_0_P_i_, N_1_P_i,_ and N_2_P_i_), the root morphology and growth status exhibited a more positive acquisition strategy for resources (Figure 3), which supported our second hypothesis. This may be related to the species habits of *C. kawakamii* seedlings in a long-term adaptation to the high nitrogen and low phosphorus soil environment in subtropical forests. Additionally, *C. kawakamii* seedlings are a shade-tolerant and ectomycorrhizal (EM) plant species. EM fungi can produce extracellular enzymes that decompose organic matter, thus effectively utilizing organic forms of nitrogen (N) and phosphorus (P) in the soil [56]. Excessive nitrogen input can alter the community composition of EM plant-dominated forest ecosystems’ soil fungi and reduce the relative abundance of EM plants [57,58]. This may also be one of the reasons why excessive nitrogen input is not conducive to nutrient absorption by *C. kawakamii* seedling roots. Furthermore, similar findings have been reported in studies investigating the impact of nitrogen addition on plant growth in subtropical regions, suggesting that excessive nitrogen addition encourages plants in these regions to adopt more conservative resource-use strategies [7,59]. However, a study focusing on the response of the dominant tree species Schima superba to nitrogen addition in subtropical evergreen broad-leaved forests yielded contrasting results with a focus on “conservative” [60]. The disparity observed in these studies may be attributed to variations in forest types, experimental conditions, and soil nutrients [7].

In general, root morphology and composition are adaptive traits of evolution that allow plants to overcome environmental limitations and obtain nutrients and water for their growth [50]. However, the response of root growth to nitrogen (N) and phosphorus (P) addition varies among different plant species, which is largely influenced by differences in root configuration, diameter, length, elemental content, and mycorrhizal type [61,62]. Based on previous studies, it is anticipated that conducting in-depth research on root morphology and composition will yield valuable insights into plant evolution and resource utilization in the future. Gaining an understanding of how different plant species respond to nitrogen and phosphorus addition will enhance our knowledge of plants’ capacity to adapt to environmental changes. The variations in root configuration, elemental composition, and mycorrhizal associations will remain critical aspects of future investigations. These studies contribute to unraveling the mechanisms by which plants cope with resource limitations through the modulation of root characteristics, and provide theoretical guidance for the conservation of natural ecosystems.

## 4. Materials and Methods

### 4.1. Materials Collection

Seedlings were collected from the *C. kawakamii* Natural Reserve located in Sanming City. Specifically, one-year-old *C. kawakamii* seedlings that displayed consistent growth and lacked branching were carefully chosen for further examination. The average height and average basal diameter of the seedlings were 18.4 ± 0.29 cm and 3.26 ± 0.059 mm, respectively. The soil used in the experiment was obtained from a depth of 0–50 cm in the reserve. The soil was screened to remove larger stones, plant roots, and other impurities. Then, the soil was uniformly mixed with carbendazim and disinfected, and fumigated by sunlight. The initial physicochemical properties of the soil are shown in Appendix A of the Appendix A.

The seedling cultivation experiment was conducted in early February 2021 in the seedling nursery of the *C. kawakamii* Natural Reserve. The selected *C. kawakamii* seedlings were transplanted into seedling pots (24 cm × 16 cm × 24 cm) and placed under a shade net (shading rate: 80%) for acclimation for two months. Each pot was filled with 4.5 kg of soil, and watering of 200 mL per pot was performed every two days to ensure sufficient moisture supply during the seedling growth stage. After the acclimation period, the seedlings were labeled for subsequent nitrogen–phosphorus addition experiments.

### 4.2. Experimental Design

According to the nitrogen deposition level in Sanming City, Fujian Province (3.6 g m^−2^ a^−1^) [21,63], nitrogen addition concentrations were set using an arithmetic progression (1, 4, 7). The low nitrogen concentrations (N_1_), medium nitrogen concentrations (N_2_), and high nitrogen concentrations (N_3_) were set at 3.6 g m^−2^ a^−1^, 14.4 g m^−2^ a^−1^, and 25.2 g m^−2^ a^−1^, respectively. Based on the local soil nitrogen-to-phosphorus ratio (6:1) [36,64], we set the low phosphorus concentrations (P_1_) at 0.6 g m^−2^ a^−1^, medium phosphorus concentrations (P_2_) at 2.4 g m^−2^ a^−1^, and high phosphorus concentrations (P_3_) at 4.2 g m^−2^ a^−1^. A total of 16 interactive treatments were set based on the nitrogen–phosphorus addition concentrations (N_0_P_0_, N_0_P_1_, N_0_P_2_, N_0_P_3_, N_1_P_0_, N_1_P_1_, N_1_P_2_, N_1_P_3_, N_2_P_0_, N_2_P_1_, N_2_P_2_, N_2_P_3_, N_3_P_0_, N_3_P_1_, N_3_P_2_, and N_3_P_3_), with 3 replicates per treatment, resulting in a total of 48 pots. The N_0_P_0_ treatment (non-nitrogen phosphorus addition treatment) was regarded as the control group (Figure 5). To facilitate analysis, we abbreviated the single phosphorus addition treatment as N_0_P_i_ treatment (where i = 1, 2, 3), the single nitrogen addition as N_i_P_0_ treatment (where i = 1, 2, 3), the low nitrogen and phosphorus interactive addition as N_1_P_i_ treatment (where i = 1, 2, 3), and the medium and high nitrogen and phosphorus interactive addition were regarded as N_2_P_i_ and N_3_P_i_, respectively. Ammonium chloride (NH_4_Cl) (analytical reagent, the relative molecular mass of N is 26.2%) was used for nitrogen addition, and sodium dihydrogen phosphate (NaH_2_PO_4_) (analytical reagent, the relative molecular mass of P is 25.82%) was used for phosphorus addition. The required mass of nitrogen and phosphorus addition was calculated based on the relative molecular masses of nitrogen and phosphorus nutrient samples and the diameter of the soil surface in the seedling pots (Appendix A).

From April to August 2021, nitrogen and phosphorus addition experiments on seedlings were conducted. Before each addition of nitrogen and phosphorus elements, the samples were dissolved in water and then added to the seedling pots. To prevent damage to the roots of seedlings caused by excessive nitrogen and phosphorus concentrations, the nitrogen and phosphorus nutrient samples dissolved in water were divided into six equal parts and added every 20 days. Before each nitrogen and phosphorus addition experiment, weeds in and around the seedling pots were removed. The positions of seedling pots were moved monthly to ensure consistent light conditions during the seedling growth period. After the nitrogen and phosphorus addition experiments, an additional buffer period of two months was allowed to ensure the sufficient absorption of nitrogen and phosphorus nutrients by the seedlings.

### 4.3. Determination of Seedling Root Traits

The seedlings were harvested in October 2021. We selected three *C. kawakamii* seedlings with consistent growth and healthy leaves from each treatment, and quickly separated the root tissue of the seedlings. The roots were cleaned of debris and soil particles and placed in a sealed plastic bag, which was then stored in a fresh-keeping box (4 °C) and transported to the laboratory. We cleaned the roots with deionized water, and wiped off the surface water of root tissue using filter paper. Furthermore, we measured the fresh weight of roots using an electronic balance (accuracy 0.0001 g). Then, the roots were scanned using an EPSON root scanner, and the root morphology was quantitatively analyzed using root image analysis software (Win Rhizo Pro 2007, Regent Instruments Inc., Quebec City, QC, Canada) to record the total root length, root surface area, root volume, root average diameter, and root branching number. Next, the scanned roots were oven-dried at 105 °C for 30 min and then dried at 65 °C to a constant weight. The root biomass was weighed using an electronic balance (accuracy 0.0001 g). The specific root length was calculated as the total root length divided by the root biomass, and the root tissue density was calculated as the root biomass divided by the root volume. The dried roots were crushed by mortar and sieved through 0.15 mm mesh. Root nitrogen content was measured using an automatic element analyzer (Vario Macro Cube, Elementar, Hanau, Germany). Root phosphorus content was determined using an inductively coupled plasma optical emission spectrometer (ICP-OES, PEOPTIMA 8000, PerkinElmer, Waltham, MA, USA). 

### 4.4. Data Analysis

For all measured variables, a normality test (Shapiro–Wilk test, α = 0.05) was conducted to assess the distribution of values, and log transforms were applied if necessary. A one-way analysis of variance (ANOVA) was performed to test the effects of different nitrogen and phosphorus additions on various traits of seedling roots. Pearson correlation coefficients were used to evaluate the bivariate correlations among root indicators. To assess the adaptive strategies of roots under different nitrogen and phosphorus additions, principal component analysis (PCA) was conducted using the “FactoMineR”, “corrplot”, and “factoextra” packages. To better differentiate the effects of different levels of nitrogen and phosphorus additions on the 16 roots characteristics, hierarchical cluster analysis was performed using the packages “Cluster”, “factoextra”, and “Stats” based on the similarities of root features. All statistical analyses were carried out using R version 4.3.2, and data plots were generated using the “ggplot2” package. A significance level of *p* < 0.05 was considered for all analyses.

## 5. Conclusions

Assessing the response of root traits of seedlings to nitrogen and phosphorus concentrations is of great significance in understanding the adaptive strategies of plants in severely phosphorus-deficient soils in the subtropical forests. The results indicated significant differences in the effects of nitrogen and phosphorus additions on seedling root traits. Under conditions of single phosphorus addition, low nitrogen and phosphorus interaction, and medium nitrogen and phosphorus interaction, the roots tend to adopt a strategy of rapid soil resource acquisition. However, under conditions of no nitrogen and phosphorus addition and single nitrogen addition, the roots tend to adopt a conservative resource-use strategy. In higher nitrogen addition concentrations, seedlings appear to adopt conservative resource-use strategies of roots. These findings reveal that seedling roots adopt a more conservative resource-use strategy in high nitrogen and low phosphorus soils, while the addition of nitrogen and phosphorus alters the resource-use strategies of seedling roots. To further protect the rare tree species within the protected area, more attention needs to be given to the underground adaptive strategies of plants and their responses to different nitrogen and phosphorus additions.

## Figures and Tables

**Figure 1 plants-13-00536-f001:**
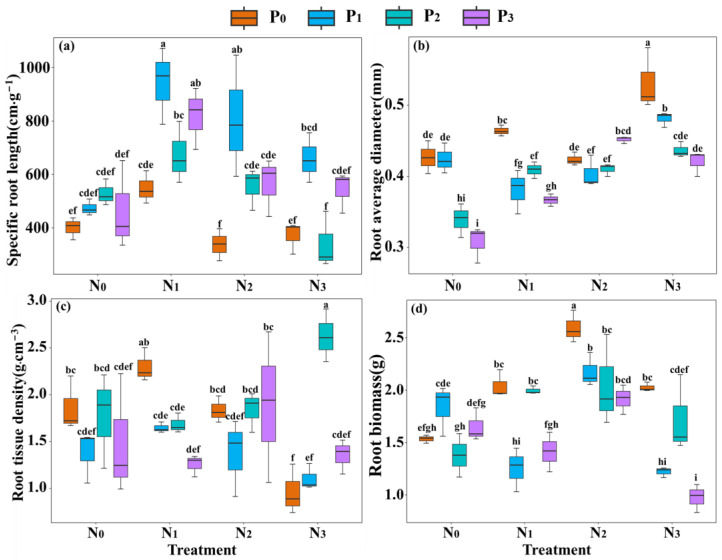
Root morphological traits of *C. kawakamii* seedlings. Note: (**a**–**d**) represent the responses of specific root length, average root diameter, root tissue density, and root biomass to nitrogen and phosphorus addition, respectively. N_0_P_i_: single phosphorus addition treatment, N_i_P_0_: single nitrogen addition treatment, N_1_P_i_: low nitrogen and phosphorus interaction treatment, N_2_P_i_: medium nitrogen and phosphorus interaction treatment, N_3_P_i_: high nitrogen and phosphorus interaction treatment, where i = 1, 2, 3. Different lowercase letters indicate significant differences (*p* < 0.05).

**Figure 2 plants-13-00536-f002:**
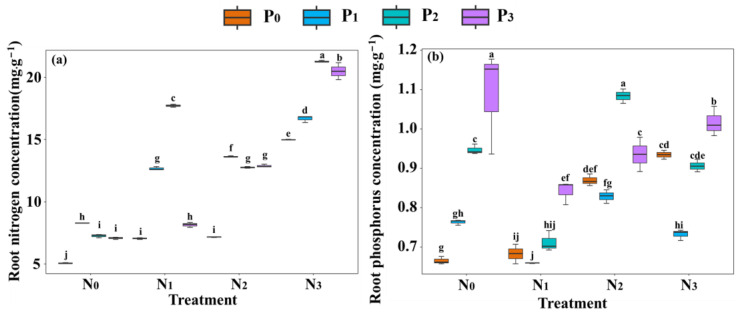
Root nitrogen and phosphorus content of *C. kawakamii* seedling. Note: (**a**) The effect of different nitrogen and phosphorus additions on root nitrogen content; (**b**) the effect of different nitrogen and phosphorus additions on root phosphorus content. Different lowercase letters indicate significant differences (*p* < 0.05).

**Figure 3 plants-13-00536-f003:**
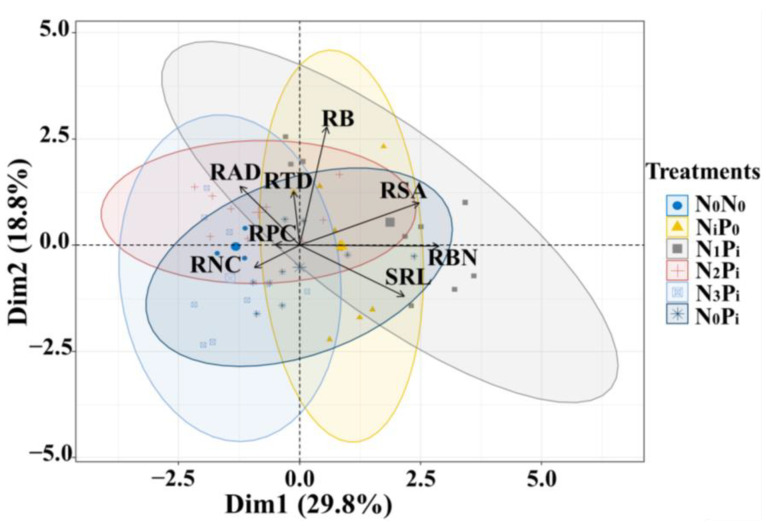
Principal component analysis (PCA) of root traits at different nitrogen and phosphorus treatments. Note: SRL, RSA, RBN, RAD, RTD, RB, RPC, and RNC represent the specific root length, root surface area, root branching number, root average diameter, root tissue density, root biomass, root phosphorus content, and root nitrogen content, respectively. N_0_P_0_ indicates a non-nitrogen phosphorus addition treatment, N_0_P_i_ indicates a single phosphorus addition treatment; N_i_P_0_ indicates a single nitrogen addition treatment; N_1_P_i_ indicates low nitrogen and phosphorus interactive addition treatment; N_2_P_i_ indicates interactive addition of medium nitrogen and phosphorus; N_3_P_i_ indicates high nitrogen and phosphorus interactive addition treatment, where i = 1, 2, 3. Different colors represent different treatments. The ellipse represents the distribution range (95% confidence interval) of each treatment.

**Figure 4 plants-13-00536-f004:**
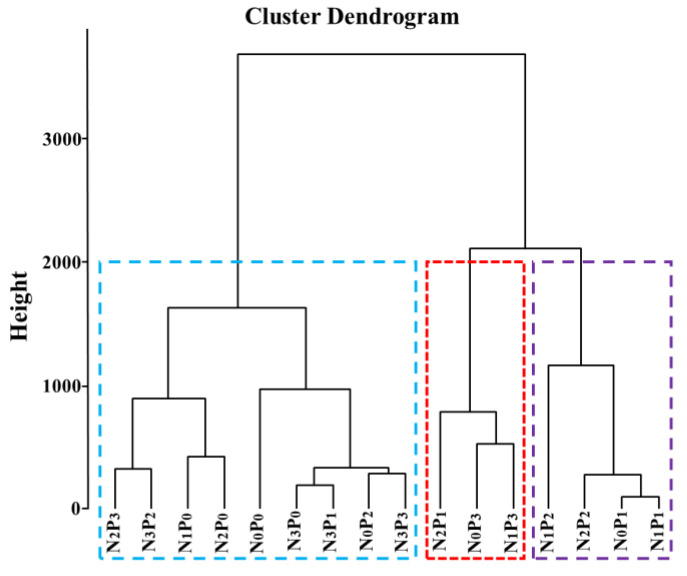
Cluster analysis of different nitrogen and phosphorus treatments. Note: The nitrogen and phosphorus addition treatments contained in the boxes indicate that they have high similarity and are aggregated together. The three boxes represent the three groups of clustering analysis results. Different colors are only used to better distinguish the three groups.

**Figure 5 plants-13-00536-f005:**
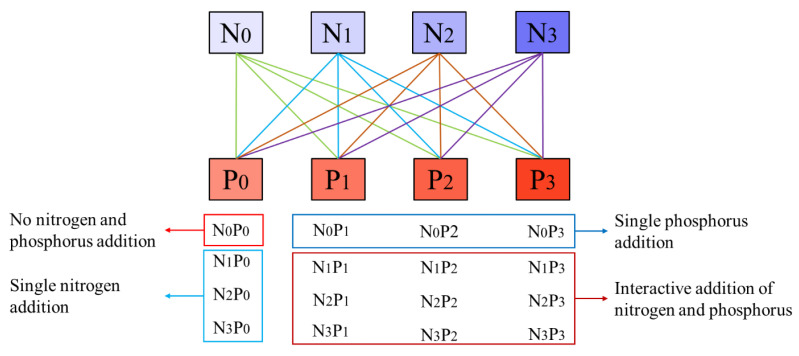
A conceptual figure of experimental design. Note: The upper section of the diagram displays an interactive representation of nitrogen and phosphorus concentrations. The colors deepen gradually from left to right, indicating an incremental increase in the concentration of nitrogen or phosphorus. The lines connect the different combinations of nitrogen and phosphorus concentrations. Distinct colored lines are used to differentiate between combinations with the same nitrogen concentration but four different phosphorus concentrations, and vice versa. The lower section demonstrates the results of an orthogonal design experiment, with 16 combinations of nitrogen and phosphorus. The combination no nitrogen and phosphorus addition (N_0_P_0_) are displayed in the red box. The light blue box contains the three combinations (N_1_P_0_, N_2_P_0_, N_3_P_0_) with a single nitrogen addition. The blue box represents the three combinations (N_0_P_1_, N_0_P_2_, N_0_P_3_) with a single phosphorus addition. Within the dark red box, nine combinations (N_1_P_1_, N_1_P_2_, N_1_P_3_, N_2_P_1_, N_2_P_2_, N_2_P_3_, N_3_P_1_, N_3_P_2_, N_3_P_3_) demonstrate the interaction between nitrogen and phosphorus additions.

## Data Availability

Data will be made available on request.

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
