# Peer review of "Adaptation Strategies of Seedling Root Response to Nitrogen and Phosphorus Addition"

_plants, 2024, doi:10.3390/plants13040536_

Round 1

Reviewer 1 Report

Comments and Suggestions for Authors

In this study, the authors explored the impact of adaptation strategies in seedling roots responding to nitrogen and phosphorus additions. The authors evaluated the response of seedling root traits to nitrogen and phosphorus addition. The study found that adding phosphorus and nitrogen-phosphorus increased specific root length, surface area, and root phosphorus content, while nitrogen addition increased average root diameter. Non-nitrogen phosphorus and single nitrogen additions tended to adopt a conservative resource-use strategy for growth under low phosphorus conditions. Single phosphorus and interactive phosphorus-nitrogen additions prompted roots to adopt an acquisitive resource-use strategy to obtain more available phosphorus resources. Accordingly, the adaptation strategy of seedling roots can be regulated by adding appropriate concentrations of nitrogen or phosphorus, promoting the natural regeneration of subtropical forests.

The work presented in this paper is commendably well-organized and well-written, providing valuable insights. However, a few minor suggestions are proposed for further enhancement. In the introduction, a clearer identification of knowledge gaps and their connection to the research questions would strengthen the paper's foundation. Emphasizing the novelty and relevance of the study at the outset will engage readers and set the stage for the subsequent findings.

Regarding Figure 1 and Figure 2, it is advisable to avoid using a red-green color combination and maintain consistent font size and formatting throughout the main text. In the discussion section, placing a stronger emphasis on highlighting insights gained from the findings and their applicability to future research is recommended. Addressing key questions, such as the research gaps addressed, the beneficiaries of the improvements, and potential future directions, will significantly enhance the discussion and underscore the paper's contribution.

Overall, this paper makes a valuable contribution to the field, and addressing these minor points will further enhance its clarity and impact.

Comments on the Quality of English Language

The overall language is commendable, demonstrating a strong grasp of the subject matter. However, there are opportunities for improvement in terms of grammar. Some sentences could benefit from restructuring to enhance the overall flow. 

Author Response

Reviewer 1

  1. The work presented in this paper is commendably well-organized and well-written, providing valuable insights. However, a few minor suggestions are proposed for further enhancement. In the introduction, a clearer identification of knowledge gaps and their connection to the research questions would strengthen the paper's foundation. Emphasizing the novelty and relevance of the study at the outset will engage readers and set the stage for the subsequent findings.

Reply: Thank you for providing us with your valuable suggestions. We agree that providing a clearer identification of knowledge gaps and the connection to the research questions would strengthen the paper's introduction. Please see the revision manuscript.

L43-46:

However, there are still many unresolved mysteries regarding how plant roots perceive and respond to changes under different soil nutrient conditions [7]. Therefore, further investigation is needed to elucidate the specific mechanisms and factors that influence this adjustment process.

L60-64:

The concealment and complexity associated with this phenomenon present significant challenges to our understanding of how root tissues adapt to soil nutrients. Thus, the elucidation of resource use strategies employed by plant roots in diverse environments holds great significance in comprehending the adaptive mechanisms of plant growth.

  1. Regarding Figure 1 and Figure 2, it is advisable to avoid using a red-green color combination and maintain consistent font size and formatting throughout the main text.

Reply: We sincerely appreciate your valuable suggestions. We agree that using a red-green color combination in Figure 1 and Figure 2 may not be the best choice, as it can pose difficulties for readers with color vision deficiencies. It is important to ensure accessibility and readability for all readers. In addition to that, maintaining a consistent font size and formatting throughout the main text is also crucial for a professional and cohesive presentation. Consistency in these aspects helps to create a visually pleasing and easy-to-follow reading experience for the audience. We revise the figures to use a more suitable and inclusive color combination, ensuring that all readers can interpret the information presented accurately. We also ensure that the font size and formatting are consistent throughout the main text, providing a cohesive and professional appearance. Please refer to the revised figure 1 and figure 2.

L154-163:

Figure 1. Root morphological traits of C. kawakamii seedlings. Note: (a), (b), (c), and (d) represent the responses of specific root length, average root diameter, root tissue density, and root biomass to nitrogen and phosphorus addition, respectively. N0Pi: single phosphorus addition treatment, NiP0: single nitrogen addition treatment, N1Pi: low nitrogen and phosphorus interaction treatment, N2Pi: medium nitrogen and phosphorus interaction treatment, N3Pi: high nitrogen and phosphorus interaction treatment, where i=1,2,3. The differences represented by different lowercase letters between treatments with different nitrogen and phosphorus concentrations added are significant (p<0.05). The same applies to the following.

L172-174:

Figure 2. Root nitrogen and phosphorus content of C. kawakamii seedling. Note: (a) The effect of different nitrogen and phosphorus additions on root nitrogen content; (b) The effect of different nitrogen and phosphorus additions on root phosphorus content.

  1. In the discussion section, placing a stronger emphasis on highlighting insights gained from the findings and their applicability to future research is recommended. Addressing key questions, such as the research gaps addressed, the beneficiaries of the improvements, and potential future directions, will significantly enhance the discussion and underscore the paper's contribution.

Reply: We appreciate your suggestions and have made the necessary modifications to address the concerns raised. Specifically, in response to your comment about the discussion section, we have revised it to emphasize the insights gained from the research findings and their applicability to future studies. We have focused on addressing key issues such as the research gap being filled, the beneficiaries of the improvements, and potential future directions. Firstly, we have conducted a thorough analysis of our research results to identify key insights derived from our study. We have considered the importance and novelty of these insights within the existing research literature. Secondly, we have linked our findings to future research by highlighting the research gap our study addresses. We emphasized the significance of this research gap in the academic field and provided insights and applications for future studies. This helps underscore the value and contribution of our research. Furthermore, we have mentioned potential future research directions. We have explored related questions or alternative methodologies that were not covered in our study, suggesting potential areas for further investigation and highlighting the potential impact of these directions. Overall, these revisions in the discussion section have allowed us to provide a more comprehensive and detailed account of the insights gained from our research. Thank you for your guidance and support throughout the review process. We believe our revised manuscript now effectively addresses your concerns and greatly improves the overall quality of the discussion section. Please see the revision manuscript.

L253-261:

In contrast, when nitrogen was applied as a single dose, it resulted in a decrease in specific root length and the number of root branches (Figure 1a and Figure S1b). This could be attributed to alterations in root allocation strategies triggered by excessive nitrogen saturation [43]. Previous research has demonstrated that an increase in soil nutrient availability often prompts plants to reduce their investment in root nutrient uptake and allocate more resources towards above-ground growth [44]. These findings suggest that the proliferation and growth of the root system can be regulated by manipulating nutrient inputs, but the precise mechanisms underlying this regulation require further investigation [7].

L288-296:

This finding, namely the synergistic effect resulting from the combined addition of nitro-gen and phosphorus, has been supported by previous studies [26, 43]. However, the precise mechanisms underlying these synergistic effects remain poorly understood. Schleiss et al. [26] propose that this mechanism can be explained by the enhanced accumulation of organic phosphorus in the soil due to the combination of nitrogen and phosphorus, as well as the stimulation of organic phosphorus hydrolysis through an increase in phosphatase activity. Furthermore, the combined addition of nitrogen and phosphorus has been shown to increase the relative abundance of arbuscular mycorrhizal fungi (AMF) genes and enhance phosphorus uptake [26].

L358-366:

Based on previous studies, it is anticipated that conducting in-depth research on root morphology and composition will yield valuable insights into plant evolution and re-source utilization in the future. Gaining understanding of how different plant species respond to nitrogen and phosphorus addition will enhance our knowledge of plants' capacity to adapt to environmental changes. The variations in root configuration, elemental composition, and mycorrhizal associations will remain critical aspects of future investigations. These studies contribute to unraveling the mechanisms by which plants cope with resource limitations through the modulation of root characteristics, and provide theoretical guidance for the conservation of natural ecosystems.

  1. Overall, this paper makes a valuable contribution to the field, and addressing these minor points will further enhance its clarity and impact.

Reply: We appreciate your suggestions to enhance the clarity and impact of the paper. Thank you once again for your feedback and your kind suggestions.

  1. Comments on the Quality of English Language. The overall language is commendable, demonstrating a strong grasp of the subject matter. However, there are opportunities for improvement in terms of grammar. Some sentences could benefit from restructuring to enhance the overall flow.

Reply: Thank you for your valuable feedback on my paper. We take note of your suggestion regarding grammar improvement. We will carefully review the paper and make necessary adjustments to address any grammar issues that may affect the clarity and coherence of the sentences. Additionally, we will pay closer attention to the structure of the sentences to ensure a better flow throughout the paper.

Reviewer 2 Report

Comments and Suggestions for Authors

 The ratio of nitrogen to phosphorus is very important for plant growth in general. Ultimately, Sprengel and Justus von Liebig's law of the minimum applies here. Unfortunately, this well-known fact is not mentioned in the introduction. This does not mean that the experiment presented is insignificant.
Unfortunately, the introduction also fails to explain the consequences of the question for the diversity of biotic communities. The authors are not dealing with that question. The focus is on the rare tree species only!

"Currently, the diversity of plant communities in the region expresses a decreasing trend due to the limitation of phosphorus in forest soil, but whether seedlings and adult trees have the same sensitivity to soil nitrogen and phosphorus elements remains poorly understood."

This explanation in the text must be discussed more in detail, especially the relation between nitrogen availability and species richness!

The methods have been well described, but the number of plants analysed per test variant is extremely low!

Please note that minor printing errors have crept in, e.g. p10, 322.

In general the introduction must be improved!

Author Response

Reviewer 2

  1. The ratio of nitrogen to phosphorus is very important for plant growth in general. Ultimately, Sprengel and Justus von Liebig's law of the minimum applies here. Unfortunately, this well-known fact is not mentioned in the introduction. This does not mean that the experiment presented is insignificant.

Reply: We appreciate your insights on the importance of nitrogen phosphorus ratio in plant growth and the correlation between Sprengel and Justus von Liebig’s minimum law. The nitrogen phosphorus ratio is crucial for plant growth, which is correct, and the minimum value law can help explain the plant's response to nutrient supply. These are important concepts that provide a basis for the significance of the experiments proposed in the paper. We will include this section in the revised manuscript.

L70-86:

With the intensification of nitrogen deposition, the accumulation of nitrogen elements in the soil can cause nitrogen-favoring species to grow rapidly in communities, enhancing their relative competitive advantage over other species and leading to changes in community structure [21]. Additionally, the long-term negative effect of reduced nitrogen forms (ammonia and ammonium) hinders the development of roots and buds. The soil mediated acidification effect increases the loss of alkaline ions, increases the concentration of potential toxic metals (such as AI3+), and reduces nitrification, leading to the accumulation of litter. The secondary stress and interference factors generated from this can also reduce individual vitality and increase herbivorous activity [22]. In summary, these factors may collectively lead to a decrease in community species diversity. Therefore, excessive nitro-gen input exacerbated the inhibitory effect of soil phosphorus limitation on root growth, which was unfavorable for roots to acquire soil resources [23]. The Sprengel and Justus von Liebig's law of the minimum has indicated that plant growth will be determined by the scarcest resources (limiting factors) [24]. According to this law, excessive nitrogen input leads to a scarcity of phosphorus, which becomes a bottleneck for plant growth. However, plants have also evolved corresponding strategies through long-term adaptation.

  1. Unfortunately, the introduction also fails to explain the consequences of the question for the diversity of biotic communities. The authors are not dealing with that question. The focus is on the rare tree species only

Reply: Thank you for expressing your concern regarding the consequences of the research question for the diversity of biotic communities. While we understand your viewpoint, we would like to clarify that the primary focus of our study is indeed on rare tree species and their response to soil nutrient availability. As such, we did not specifically address the implications for the overall diversity of biotic communities in our research. However, we appreciate your suggestion of incorporating a discussion on the broader ecological implications. While it may not be within the scope of this particular study, considering and acknowledging the potential indirect effects on the diversity of biotic communities could be a valuable aspect to explore in future research. we apologize if the introduction did not meet your expectations in addressing this aspect, but we hope you understand the limitations and specific focus of our study on rare tree species. Nonetheless, your feedback is valuable, and we will take it into account for future studies to provide a more comprehensive understanding of the relationships between soil nutrients and biodiversity. Thank you for sharing your thoughts, and we appreciate your contribution to the academic discourse.

  1. "Currently, the diversity of plant communities in the region expresses a decreasing trend due to the limitation of phosphorus in forest soil, but whether seedlings and adult trees have the same sensitivity to soil nitrogen and phosphorus elements remains poorly understood." This explanation in the text must be discussed more in detail, especially the relation between nitrogen availability and species richness

Reply: Thank you for your comment. We understand your request in the text to further discuss the relationship between nitrogen availability and species richness. In addition, we acknowledge that upon careful examination, we have found that this study did not involve any content related to the response of adult trees to nitrogen and phosphorus addition. We apologize for any confusion that may have been caused by previous modifications. In order to discuss in more detail, the relationship between nitrogen availability and species richness, we also further explore existing literature on this topic to review research on the impact of nitrogen availability on plant community species richness, and discuss the potential mechanisms driving this relationship. This discussion helps to gain a deeper understanding of the potential impact of nitrogen availability on plant community species composition and overall diversity. By combining these additional details and discussing the relationship between nitrogen availability and species richness, this article will have a more comprehensive understanding of the topic. Please see the revision manuscript.

L73-86:

Additionally, the long-term negative effect of reduced nitrogen forms (ammonia and ammonium) hinders the development of roots and buds. The soil mediated acidification effect increases the loss of alkaline ions, increases the concentration of potential toxic metals (such as AI3+), and reduces nitrification, leading to the accumulation of litter. The secondary stress and interference factors generated from this can also reduce individual vitality and increase herbivorous activity [22]. In summary, these factors may collectively lead to a decrease in community species diversity. Therefore, excessive nitrogen input exacerbated the inhibitory effect of soil phosphorus limitation on root growth, which was unfavorable for roots to acquire soil resources [23]. The Sprengel and Justus von Liebig's law of the minimum has indicated that plant growth will be determined by the scarcest resources (limiting factors) [24]. According to this law, excessive nitrogen input leads to a scarcity of phosphorus, which becomes a bottleneck for plant growth. However, plants have also evolved corresponding strategies through long-term adaptation.

L108-117:

Currently, the diversity of plant communities in the region expresses a decreasing trend due to the limitation of phosphorus in forest soil, but whether seedlings have the same sensitivity to soil nitrogen and phosphorus elements remains poorly understood. As previously mentioned, nitrogen deposition can induce a series of processes that can result in a decline in the diversity of community species [22]. This effect is particularly noteworthy in acidic soils in subtropical regions of China [3]. In such soils, phosphorus tends to persist in an insoluble form through adsorption and fixation with substances like iron, aluminum, and soil clay [5]. Consequently, the conversion of inorganic phosphorus into insoluble phosphorus exacerbates the restriction of soil phosphorus on plant growth [3,5].

  1. The methods have been well described, but the number of plants analysed per test variant is extremely low!

Reply: Thank you for your valuable comments regarding the number of plants analyzed for each test variant in our study. We appreciate your concern regarding the potential impact of low sample sizes on the statistical power and generalizability of our research findings. We acknowledge the limitations associated with small sample sizes and understand the importance of considering the specific background and limiting factors of our study. We recognize that logistical constraints and limited availability of resources might have constrained our ability to conduct extensive field experiments. Nevertheless, we fully recognize the significance of sample size and its influence on statistical significance. While accounting for any limitations or challenges we may encounter, we agree that future research should strive for more detailed grouping, larger sample sizes, and deeper investigations to obtain more accurate and reliable outcomes. We thank you for bringing this issue to our attention, and we assure you that we will incorporate this feedback into our future research endeavors.

  1. Please note that minor printing errors have crept in, e.g. p10, 322.

Reply: Thank you for bringing the minor printing error in our paper to our attention. We highly appreciate your careful review and your diligence in pointing out these errors. We fully recognize the significance of accuracy and precision in academic writing and sincerely apologize for any inconvenience caused. Please rest assured that we will thoroughly review the paper to identify and correct any possible typing errors, missing information, or other printing errors. Furthermore, we will conduct a comprehensive review of the entire manuscript to ensure that any similar errors are addressed and resolved. Please see the revision manuscript.

L333-337:

In contrast, under the treatment of single phosphorus addition, low nitrogen and medium nitrogen and phosphorus interactive addition (N0Pi, N1Pi and N2Pi), the root morphology and growth status exhibited a more positive acquisition strategy for resources (Figure 3), which supported our second hypothesis.

L339-341:

Additionally, C. kawakamii seedlings are shade tolerant and ectomycorrhizal (EM) plant species, EM fungi can produce extracellular enzymes that decompose organic matter, thus effectively utilizing organic forms of nitrogen (N) and phosphorus (P) in the soil [56].

L369-371:

Seedlings were collected from the C. kawakamii Natural Reserve located in Sanming City. Specifically, one-year-old C. kawakamii seedlings that displayed consistent growth and lacked branching were carefully chosen for further examination.

  1. In general the introduction must be improved!

Reply: Thank you for your insightful opinions and suggestions. We will handle each point accordingly:

  1. We appreciate your feedback on the importance of nitrogen phosphorus ratio for plant growth and the application of Sprengel and Justus von Liebig’s minimum law. We agree that this is an important aspect that needs to be considered and should be mentioned in the introduction.
  2. We understand your concerns about the impact of research questions on biodiversity, and we appreciate your suggestion to include discussions on broader ecological impacts. We apologize if the introduction did not meet your expectations in addressing this issue, but we hope you understand the limitations and specific focus of our research on rare tree species.
  3. We appreciate your comments on the explanation of the relationship between nitrogen availability and species richness in the text. I agree that further discussion is needed on this aspect, especially the connection between nitrogen availability and its impact on species richness. We ensure that this relationship is elaborated in detail in the text to better understand the research background. Please see the revision manuscript.

Reviewer 3 Report

Comments and Suggestions for Authors

Article present the adaptation strategies of seedlings' roots in response to nitrogen and phosphorus addition 

The article presents interesting topics. It is poorly formatted. The authors present an interesting study. However, in the discussion they were too much in discussing the results of the research. There is a lack of references to other studies.

 You've cited studies that both support and contradict your findings. How do you reconcile these differences? Could they be due to differences in experimental conditions, species used, or soil types? 

Author Response

Reviewer 3

  1. The article presents interesting topics. It is poorly formatted. The authors present an interesting study. However, in the discussion they were too much in discussing the results of the research. There is a lack of references to other studies.

Reply: We appreciate your acknowledgement of the interesting topics presented. We apologize for any formatting issues that may have affected the readability of the article. Regarding the discussion section, we understand your concern regarding an excessive focus on the results of our research. We acknowledge that the inclusion of references to other relevant studies would greatly enhance the strength of our discussion. We will address this oversight by thoroughly reviewing the literature and incorporating appropriate references to support and contextualize our findings. We value your input and strive to improve the quality of our work. We appreciate your highlighting these aspects and will make the necessary revisions to provide a more comprehensive and well-supported discussion.

L235-244:

These discrepancies in the response of root growth to nitrogen (N) and phosphorus (P) additions observed between different plant species can likely be attributed to variations in experimental conditions and soil types [7]. The study area in our research is characterized by subtropical evergreen broad-leaved forests, which are known to be relatively deficient in phosphorus [7, 49]. In contrast, the study area of latter's research focuses on tropical lowland forests, which are typically more fertile in terms of phosphorus. Furthermore, the type of phosphorus fertilizer used in their study (superphosphate) differs from the one used in our research (sodium dihydrogen phosphate) [41]. These differences in soil com-position and phosphorus fertilizer application can contribute to the varying responses observed in root growth between the two studies.

L250-261:

Our findings align with those of previous studies, which demonstrate that the addition of nitrogen facilitates passive absorption by the root system. This process leads to an increase in root diameter, allowing for greater nutrient storage to meet the demands of passive ab-sorption [43,44]. In contrast, when nitrogen was applied as a single dose, it resulted in a decrease in specific root length and the number of root branches (Figure 1a and Figure S1b). This could be attributed to alterations in root allocation strategies triggered by excessive nitrogen saturation [43]. Previous research has demonstrated that an increase in soil nutrient availability often prompts plants to reduce their investment in root nutrient uptake and allocate more resources towards above-ground growth [44]. These findings suggest that the proliferation and growth of the root system can be regulated by manipulating nutrient inputs, but the precise mechanisms underlying this regulation require further investigation [7].

L285-296:

This indicates that nitrogen-phosphorus interaction not only significantly promotes the growth, elongation, and branching differentiation of seedling roots but also has a greater promoting effect compared to the same level of single nitrogen or phosphorus addition. This finding, namely the synergistic effect resulting from the combined addition of nitro-gen and phosphorus, has been supported by previous studies [26, 43]. However, the precise mechanisms underlying these synergistic effects remain poorly understood. Schleiss et al. [26] propose that this mechanism can be explained by the enhanced accumulation of organic phosphorus in the soil due to the combination of nitrogen and phosphorus, as well as the stimulation of organic phosphorus hydrolysis through an increase in phosphatase activity. Furthermore, the combined addition of nitrogen and phosphorus has been shown to increase the relative abundance of arbuscular mycorrhizal fungi (AMF) genes and enhance phosphorus uptake [26].

  1. You've cited studies that both support and contradict your findings. How do you reconcile these differences? Could they be due to differences in experimental conditions, species used, or soil types?

Reply: Thank you for your suggestion. Recognizing and addressing these differences is crucial for ensuring a comprehensive understanding of the research topic. The changes observed in the results of different studies can indeed be attributed to several factors, including basically the same in experimental conditions, including the species and soil type in this experiment. In our research, we strive to control the experimental conditions to ensure consistency. Please see the revision manuscript.

L235-244:

These discrepancies in the response of root growth to nitrogen (N) and phosphorus (P) additions observed between different plant species can likely be attributed to variations in experimental conditions and soil types [7]. The study area in our research is characterized by subtropical evergreen broad-leaved forests, which are known to be relatively deficient in phosphorus [7, 49]. In contrast, the study area of latter's research focuses on tropical lowland forests, which are typically more fertile in terms of phosphorus. Furthermore, the type of phosphorus fertilizer used in their study (superphosphate) differs from the one used in our research (sodium dihydrogen phosphate) [41]. These differences in soil composition and phosphorus fertilizer application can contribute to the varying responses observed in root growth between the two studies.

L345-352:

Furthermore, similar findings have been reported in studies investigating the impact of nitrogen addition on plant growth in subtropical regions, suggesting that excessive nitrogen addition encourages plants in these regions to adopt more conservative resource-use strategies [7,59]. However, a study focusing on the response of the dominant tree species Schima superba to nitrogen addition in subtropical evergreen broad-leaved forests yielded contrasting results with a focus on "conservative" [60]. The disparity observed in these studies may be attributed to variations in forest types, experimental conditions, and soil nutrient [7].
